# Correlation between Mild Traumatic Brain Injury-Induced Inflammatory Cytokines and Emotional Symptom Traits: A Systematic Review

**DOI:** 10.3390/brainsci12010102

**Published:** 2022-01-12

**Authors:** Shazia Malik, Omar Alnaji, Mahnoor Malik, Teresa Gambale, Michel Piers Rathbone

**Affiliations:** 1Neurosciences Graduate Program, McMaster University, Hamilton, ON L8S 4L8, Canada; 2Faculty of Life Sciences, McMaster University, Hamilton, ON L8S 4L8, Canada; alnajio@mcmaster.ca; 3Bachelor of Health Sciences Program, McMaster University, Hamilton, ON L8S 4L8, Canada; malikm45@mcmaster.ca; 4Division of Neurology, Department of Medicine, McMaster University, Hamilton, ON L8S 4L8, Canada; gambale@mcmaster.ca (T.G.); mrathbon@mcmaster.ca (M.P.R.)

**Keywords:** concussion, mild traumatic brain injury, inflammatory cytokines, neuroinflammation, depression, PTSD

## Abstract

Both mild traumatic brain injuries (mTBI) and systemic injuries trigger a transient neuroinflammatory response that result in similar clinical outcome. The ensuing physical, cognitive, and emotional symptoms fail to subside in approximately 15–20% of the concussed population. Emotional impairments, particularly depression, anxiety, and post-traumatic stress disorder (PTSD), are commonly associated with poor recovery following mTBI. These emotional impairments also have a significant neuroinflammatory component. We hypothesized that the inflammatory cytokines seen in mTBI patients with emotional symptoms would coincide with those commonly seen in patients with emotional symptoms without mTBI. A systematic review was conducted to identify the most common neuroinflammatory cytokines in the mTBI population with psychological symptoms (depression, anxiety, PTSD). The electronic databases EMBASE, MEDLINE, Cochrane Central Register of Controlled Trials (CENTRAL), PUBMED, and PSYCINFO were searched from data inception to 31 August 2021. A systematic screening approach was employed from screening to data analysis. A total of 994 articles were screened, 108 were selected for full article review, and 8 were selected for data analysis. The included studies consisted of 875 patients of which 81.3% were male. The mean sample size of patients with at least one mTBI was 73.8 ± 70.3 (range, 9–213), with a mean age of 33.9 ± 4.8 years. The most common cytokines associated with poor psychological outcomes involving PTSD and/or depression in the chronic mTBI population were IL-6, TNFα, IL-10, and CRP.

## 1. Introduction

Mild traumatic brain injuries (mTBI), commonly known as concussions, account for approximately 80–90% of all traumatic brain injuries (TBI) [1,2]. Following mTBI, people suffer from a myriad of physical, cognitive, and psychological/emotional symptoms, collectively referred to as post-concussive symptoms (PCS). A minority of patients, about 15–20%, recover slowly or not at all [3,4,5]. Almost all of these have depressive or anxiety symptoms [6].

Concussion acutely triggers a cascade of biomolecular changes in the brain [7]. These lead to behavioral changes amongst other symptoms following concussion. Interestingly, it was observed that patients who sustain systemic injuries not involving the brain have the same characteristic symptoms as those who suffer concussions [8]. This raised the possibility that humoral mechanisms triggered by the systemic injury produced the same effect on the brain as a concussive injury to the brain itself [9].

Both systemic and neurotrauma cause inflammatory changes, and presumably the systemic inflammatory cytokines likely affect the brain in the same way as the well-described inflammatory changes that follow concussive brain injury [9]. These changes include the activation of immune cells and increased systemic concentrations of circulating inflammatory cytokines. Immediately following a concussion, neuroinflammation seems to play a role in neuroprotection; however, continued neuroinflammation can be detrimental and could be responsible for persistent symptoms [10,11]. We have identified that each symptom of PCS could be induced by altered levels of inflammatory cytokines [9]. For example, headaches are associated with elevated Tumor Necrosis Factor-α (TNF-α), Interleukin-10 (IL-10), Interleukin-1β (IL-1β), Interleukin-6 (IL-6), and Interferon-α (IFN-α) levels [9,12,13]. Cognitive impairments are associated with upregulated IL-1β, IL-6, and TNF-αlevels [9,14]. Fatigue and sleep disturbances are also associated with cytokines TNF-α, IL-6, and IL-1β [9,15,16]. Similarly, our previous review has demonstrated that depression and anxiety are associated with elevated TNF-α, IL-6, IFN-α, and C-Reactive Protein (CRP) levels [9]. Overall, the most common systemic cytokines associated with mTBI include IL-6 [17,18,19,20,21,22,23], TNF-α [20,23,24,25], IL-10 [20,26], IL-1β [19,20], Interleukin-8 (IL-8) [20,25], Interferon Gamma (IFN-γ) [20,25], Interleukin-1RA (IL-1RA) [17,21], Interleukin-4 (IL-4) [20], and C–C motif chemokine ligand 2 (CCL2) [19,27].

Similar to concussed patients, systemic inflammation seems to be associated with psychological conditions such as depression, anxiety, and post-traumatic stress disorder (PTSD) in the non-concussed population [28,29,30]. This led us to question whether the same inflammatory mediators were active in concussed patients and played a role in the genesis of post-concussion emotional impairments, particularly depression, anxiety, and PTSD, that are hallmarks of persistent post-concussion symptoms (PPCS) [6].

To help us understand the relation between inflammation and the emotional symptoms seen after mTBI, we conducted a systematic review to identify the most common neuroinflammatory cytokines associated with poor emotional recovery in the mTBI population. This study focuses on mTBI/concussions as they are the most common among all TBI subtypes [1,2]. Additionally, emotional symptoms are more predominant in this TBI subgroup [6]. We hypothesize that the inflammatory cytokines seen in mTBI patients with emotional symptoms would coincide with those commonly seen in patients with emotional symptoms without mTBI.

## 2. Methods

### 2.1. Search Strategy

Three separate searches were conducted across five databases (PUBMED, EMBASE, MEDLINE, Cochrane Central Register of Controlled Trials (CENTRAL), and PSYCINFO) for the literature on mTBI. All three searches were identical except for the outcomes in question (depression, anxiety, and PTSD). The searches only included literature from data inception to 31 August 2021. The search terms included “mild traumatic brain injury”, “neuroinflammation”, “concussion”, and similar phrases (Appendix A). A manual search using Google Scholar was conducted to capture any articles that may have been missed. The inclusion and exclusion criteria, as well as the research question, were established a priori. Inclusion criteria were: (1) mTBI; (2) neuroinflammation or at least one blood or cerebrospinal fluid (CSF) cytokine identified for the population of interest; (3) emotional symptomatology (anxiety, depression or PTSD); (4) human studies; (5) English language. Exclusion criteria were: (1) moderate and severe TBI (Glascow Coma Scale (GCS) < 13); (2) no inflammatory markers; (3) no emotional symptomatology; (4) review articles; (5) cadaver/non-human studies. This study exclusively focused on mTBI, and excluded any studies that included only moderate or severe TBIs or did not make a distinction between various types of TBIs. The full search strategy is provided as Appendix A.

### 2.2. Study Screening

In fulfilment of the Revised Assessment of Multiple Systematic Reviews (R-AMSTAR) and the Preferred Reporting Items for Systematic Reviews and Meta-analyses (PRISMA) guidelines, a systematic screening approach was implemented [31,32]. Two independent reviewers conducted the study screening in duplicate, from title to the full-text screening stage. Any discrepancies were discussed between the reviewers and were resolved thereafter. The same systematic approach was also used to screen the references of the included studies to capture any additional relevant papers (Figure 1).

### 2.3. Data Abstraction

Two independent reviewers abstracted the relevant data from the articles included in this review. The data included the author, year of publication, sample size, study design, acute (<one month) and chronic (>one month) concussion, and patient demographics (e.g., age, gender, etc.) for each study. Data regarding cytokine levels, time of cytokine analysis, biospecimen analyzed, and screening tools for anxiety/PTSD/depression were also recorded. The reviewers documented the data onto a shared spreadsheet.

### 2.4. Quality Assessment

The methodological index for non-randomized studies (MINORS) was used to evaluate the quality of all included studies in this review [33]. Each reviewer recorded a score from 0 to 2 for each of the 12 categories on the MINORS checklist. For comparative studies, a maximum score of 24 could be achieved, whereas a maximum score of 16 was possible for non-comparative studies. For each of the included non-comparative and comparative studies, a total score of 13–16 or 19–24 was considered excellent quality, 9–12 or 13–18 was considered fair quality, and 0–8 or 0–12 was considered poor quality, respectively [33]. The levels of evidence for all included studies were also assessed.

### 2.5. Statistical Analysis

Descriptive statistics including mean, range, standard deviation, and 95% confidence interval (CI) are presented where appropriate. A kappa (*κ*) statistic was used at each screening stage to assess the level of agreement between the reviewers. Near perfect agreement was categorized as any *κ* value between 0.81 to 0.99. Furthermore, a *κ* of 0.61 to 0.80 was considered significant agreement, a *κ* of 0.41 to 0.60 was moderate agreement, a *κ* of 0.21 to 0.40 was fair agreement and a *κ* value of 0.20 or less was considered slight agreement.

## 3. Results

### 3.1. Study Characteristics

A combined total of 994 papers were yielded across the five databases for the three searches. The specific searches for depression, anxiety, and PTSD yielded 576, 216, and 202 papers, respectively. A systematic screening process was followed as shown in Figure 1, yielding eight papers that met the inclusion criteria after excluding duplicates. Of the included studies, seven were case control studies (87.5%), one was a cohort study (12.5%), and two (25.0%) were conference abstracts. The main study characteristics and outcomes are described on Table 1.

### 3.2. Study Quality

All the studies included in this review have a level of evidence of IV (*n* = 8; 100%). There was considerable agreement between the two reviewers at the title/abstract screening stage (*κ* = 0.84 [95% CI, 0.70 to 0.90]) and the full-text screening stage (*κ* = 0.75 [95% CI, 0.60 to 0.90]. The mean MINORS score for non-comparative and comparative studies were 12.0 ± 1.4 and 20.1 ± 1.8 respectively, which indicates fair quality of evidence for non-randomized studies. The areas of best performance based on the MINORS checklist were “endpoints appropriate for aim” (*n* = 8; 100%) and “inclusion of consecutive patients” (*n* = 8; 100%). The area of worst performance was “unbiased assessment of endpoints”, which was not found in any of the included studies.

### 3.3. Patient Characteristics

A total of 875 patients were included across all the studies in this review. The mean sample size of patients with at least one mTBI was 73.8 ± 70.3 (range, 9–213). Furthermore, the mean sample size for non-mTBI/healthy controls was 47.7 ± 24.4 (range, 22–82). The mean age of the study groups (at least one mTBI) was 33.9 ± 4.8 years, while the mean age of the control groups was 31.3 ± 6.3 years. In addition, 81.3% (442/544) of the participants were male; two studies did not specify the sex distribution of their population [34,35]. Of the included studies, two studies did not specify age, and two did not include a healthy control population (Table 1).

### 3.4. Outcomes

#### 3.4.1. Depression

All the included papers in this review had a study population (*n* = 875) that had symptoms of depression (Table 1). The screening tools that were utilized to diagnose depression in the study populations for each paper include the nine-item Patient Health Questionnaire Depression Scale (PHQ-9) (*n* = 2), the Quick Inventory of Depressive Symptomatology (QIDS) (*n* = 2), the Center for Epidemiologic Studies Depression Scale (CES-D) (*n* = 1), and the Beck Depression Inventory (BDI) (*n* = 1). One paper did not specify the screening tools used to diagnose depression [34]. The biomarkers assessed in the studies involving depression include IL-6 (*n* = 5), IL-10 (*n* = 4), TNF-α (*n* = 4), IL-1β (*n* = 2), CRP (*n* = 2), and IL-7 (*n* = 1). The study populations across all papers involving those diagnosed with depression were either military personnel (74.6%; *n* = 653/875), athletes (1.0%, *n* = 9/875), or admitted at the hospital (24.3%, *n* = 213/875).

#### 3.4.2. Post-Traumatic Stress Disorder (PTSD)

Six of the included papers in this review involved a study population (*n* = 653) that had symptoms of PTSD (Table 1). PTSD was assessed through the PTSD Checklist Military Version (PCL-M) (*n* = 4) as well as the Post-traumatic Stress Checklist—Civilian Form (PCL-C) (*n* = 1). One study did not specify the screening tools utilized to assess PTSD [34]. The biomarkers assessed in the studies involving PTSD include IL-6 (*n* = 5), IL-10 (*n* = 4), TNF-α (*n* = 4), IL-1β (*n* = 1), CRP (*n* = 1), and IL-7 (*n* = 1). The study populations across all papers involving those diagnosed with PTSD were all veteran military personnel (*n* = 653).

#### 3.4.3. Anxiety

Two of the included papers in this review had a study population (*n* = 222) that had symptoms of anxiety (Table 1). The Beck Anxiety Inventory (BAI) was utilized as a screening tool for anxiety in one of the studies [36]. One study did not specify the screening tools utilized to assess anxiety [35]. The relevant biomarkers assessed in the anxiety study population were CRP and IL-1β. The study populations across all papers involving those diagnosed with anxiety were all athletes (4.1%, *n* = 9) or admitted to the hospital (95.9%, *n* = 213).

#### 3.4.4. Cytokines

The most notable biomarkers assessed in the eight studies included in this systematic review were IL-6 (66.7%; *n* = 584), TNF-α (54.7%; *n* = 479), IL-10 (54.7%; *n* = 479), CRP (32.2%; *n* = 282), IL-7 (12.0%; *n* = 105), and IL-1β (9.8%; *n* = 86). Four studies assessed a biomarker that was chosen to be excluded due to a small sample size or lack of clinical significance (e.g., IL-4 and IL-17a) (Table 1). Seven studies used blood samples to measure the levels of biomarkers, while one study analyzed CSF (Table 1). All eight studies assessed cytokine levels in chronic mTBI, and two assessed cytokine levels in acute mTBI.

##### IL-6

Five of the eight included studies assessed IL-6 levels within the mTBI population (Table 1) [26,34,37,38,39]. 

Kanefsky et al. (2019) demonstrated increased prevalence of adverse emotional outcomes (PTSD *p* < 0.001; depression *p* = 0.001) in mTBI patients, in comparison to non-mTBI controls. It was also observed that chronic mTBI patients with loss of consciousness (LOC) had significantly elevated IL-6 levels compared to those without LOC and non-mTBI controls. This study controlled for factors affecting both inflammatory cytokine levels and psychiatric conditions, rendering results more reliable [38]. 

Peskind et al. (2015) and Guedes et al. (2020), did not detect any variations in plasma IL-6 levels between concussed and non-concussed military veterans. However, both studies noted increased PTSD and depression prevalence among concussed individuals compared to non-concussed individuals. Neither of the two studies controlled for inflammatory conditions [34,37]. Guedes et al. (2020) discovered a mild correlation between exosomal IL-6 levels and PTSD (*p* = 0.08), despite controlling for confounding psychiatric variables [37]. Interestingly, Peskind et al. (2015), noted that CSF IL-6 levels were upregulated in military veterans (both concussed and non-concussed) in comparison to community controls, indicating prevalence of increased inflammation among military personnel [34]. 

Gill et al. (2018) did not find discrepancies in IL-6 levels within neuronal derived exosomes between concussed and non-concussed military personnel, despite controlling for inflammatory conditions [26]. 

Vedantam et al. (2021) found significantly upregulated plasma IL-6 levels within mTBI patients when compared to orthopedic injury (OI) controls, both at 24 h (*p* = 0.01) and six months post-injury (*p* = 0.044) [39]. However, the regression model did not demonstrate any statistically significant association between elevated IL-6 levels and emotional outcomes across mTBI population. This study controlled for confounding psychiatric conditions.

##### TNF-α

In this systematic review, four of the eight included studies assessed TNF-α levels in mTBI populations [26,37,38,39]. However, none of these studies found any significant variations in systemic TNF-α levels between concussed and non-concussed groups. Two of these four studies controlled for both confounding inflammatory and psychiatric variables [3,26]. 

Guedes et al. (2020) together with Kanefsky et al. (2021) found statistically significant findings regarding TNF-α levels and adverse psychological outcome(s) within the mTBI population [37,38]. Guedes et al. (2020) found that chronically elevated plasma TNF-α levels correlated weakly with PTSD (*r* = −0.2267, *p* = 0.0255), after controlling for confounding psychiatric variables [37]. These findings were supported by Kanefsky et al. (2020), who found that there is a link between chronic mTBI and PTSD symptoms through TNF-α after controlling for factors affecting both inflammatory cytokine levels and psychiatric conditions [38].

##### IL-10

In this review, four of the eight studies assessed IL-10 levels within mTBI populations [26,37,38,39]. Out of these, two studies found a statistically significant relationship between IL-10 levels and emotional symptoms [26,39]. 

Gill et al. (2018) found elevated IL-10 levels in patients with at least one mTBI event, in comparison to healthy controls. There was also a significant relationship between upregulated IL-10 levels and PTSD (*B* = 0.8, *t* = 2.60, *p* < 0.01), and a weak relationship between IL-10 levels and depression (*B* = 0.421, *t* = 1.41, *p* = 0.063) within mTBI populations. This study controlled for factors affecting both inflammatory cytokine levels and psychiatric conditions [26]. 

These findings were supported by Vedantam et al. (2021), who showed that elevated IL-10 levels at six months post-injury were significantly correlated with depression (*p* = 0.001) and with more severe PTSD symptoms (*p* = 0.004) [39]. However, this study was unable to find significant variations in plasma IL-10 levels between mTBI patients and orthopedic injury controls, both at 24 h and six months post-injury (*p* > 0.05). The lack of discrepancies in IL-10 levels between the two populations, however, could be attributed to the ability of both mTBI and OI to cause inflammation, as well as the absence of controls for confounding inflammatory variables.

Guedes et al. (2020) and Kanefsky et al. (2021), on the other hand, did not find any significant link between IL-10 levels with mTBI and emotional symptoms [37,38].

##### CRP

Two of the eight studies that this study reviewed assessed CRP levels in mTBI populations [36,40]. 

Ghai et al. (2020) found significantly elevated CRP levels in chronic mTBI veterans in comparison to controls (non-concussed veterans and community controls) [40]. Concussed veterans also had significantly greater co-morbid PTSD and depressive symptoms, in comparison to controls. This study did not control for confounding inflammatory conditions, though datasets were controlled for psychiatric variables. 

Su et al. (2014) also found that elevated baseline CRP levels were associated with an upregulated risk of persistent PCS (2.72; 95% CI: 1.61–4.59), persistent psychological issues (1.54; 95% CI: 1.06–2.22), and persistent cognitive impairments (1.69; 95% CI: 1.14–2.51) within the mTBI population. However, no non-mTBI controls were included in this study [36].

##### IL-1β

In this review, two of the eight included studies assessed IL-1β levels in mTBI populations (Table 1) [35,39]. Bellgowan et al. (2012) revealed that IL-1β levels were significantly elevated at 24–48 h post-concussion, in comparison to either one week (*p* < 0.01) or one month post-concussion (*p* < 0.05) [35]. There was no variation in IL-1β levels between one week and one month post-concussion. It was also observed that both depressive and anxiety ratings were significantly upregulated at 24–48 h and one week (*t* [7] = –3.59; *p* < 0.01; *t* [7] = –2.51; *p* < 0.05; respectively), though not at one month post-concussion. Since there were no non-mTBI controls used in this study, no conclusion can be placed regarding how IL-1β levels in mTBI differ from those of controls.

Vedantam et al. (2021) found no significant differences between IL-1β levels, both at 24 h and at six months post-mTBI, in comparison to orthopedic controls. The relationship between IL-1β levels and emotional symptoms was not made clear in this study [39].

##### IL-7

Peskind et al. (2015) assessed CSF IL-7 in mTBI populations. In comparison to deployed control veterans, mTBI veterans had greater post-concussive symptoms, combat exposure, PTSD, depression, sleep disturbance, and alcohol use. CSF IL-7 was elevated in mTBI veterans, in comparison to deployed and community controls (15.762.7 [SEM], 8.563.2, and 8.364.4 pg/mL (*p* < 0.03) [34].

## 4. Discussion

This study found evidence supporting the association between upregulated cytokine levels (IL-6, TNF-α, IL-10, CRP, and IL-1β) and adverse psychological outcomes in mTBI patients.

Previously, our group reviewed evidence that individuals with systemic injuries developed near-identical symptoms to concussion patients [9]. Since concussions are associated with cerebral inflammatory responses [41] that are responsible for post-concussion symptoms, we hypothesized that circulating inflammatory mediators similarly affected the brain following systemic injury. According to our previous review, IL-6, IFN-α, and TNF-α and CRP were associated with depression and anxiety. Furthermore, irritability was found to be associated with TNF-α and IL-1β [9]. The results of this current review align with our previously proposed hypothesis.

IL-6 acts as a regulator of inflammatory procedures by inducing either a pro- or anti-inflammatory response [11]. IL-6 levels are typically upregulated shortly after a mTBI event [21,22], and can remain elevated for weeks or months following such an injury [18,23]. Acutely elevated IL-6 levels are associated with greater symptom duration and severity in mTBI [17,21,38]. One study, however, does not support such an association [42]. This discrepancy could be attributed to population differences, as Di Battista et al., (2020) assessed cytokine levels in both males and females whereas others did so in a predominantly male population. Elevated IL-6 levels are also seen in psychological conditions, even in the absence of head trauma. For example, PTSD patients have elevated IL-6 (42% higher, *p* = 0.02) levels that are significantly related to condition severity [43]. Similarly, significantly upregulated IL-6 levels are observed in depressed and anxious populations in comparison to controls [44,45,46,47]. In this review, only two of the five studies assessing IL-6 levels found upregulated levels in concussed patient populations compared to controls [38,39]. However, it is evident that most of the included studies assessing IL-6 did not control for the confounding inflammatory conditions or treatments that could affect IL-6 levels. Furthermore, three out of these five studies showed that mTBI populations have increased depression and PTSD symptoms in comparison to non-concussed populations, out of which only two controlled for confounding psychiatric conditions [34,37,38]. One study did not show any statistically significant relationship between elevated IL-6 levels and emotional outcome in mTBI population [39]. Overall, based on the available literature, elevated IL-6 levels are associated with PTSD in the chronic mTBI population. However, more research is needed to further explore this association.

TNF-α is a pro-inflammatory cytokine associated with the neuroinflammatory responses following an mTBI event [11]. TNF-α levels are elevated in both acute [20,25] and chronic [20,23,25] inflammation, following mTBI. In children, acutely elevated TNF-α proteomic expression at 1–4 days post-mTBI (*p* = 0.031) is associated with persisting symptoms [48]. Elevated TNF-α levels are associated with poor psychological health even in the absence of an mTBI event [49]. PTSD and depression are consistently seen to be associated with elevated TNF-α levels, in comparison to healthy controls [50,51,52,53,54,55]. Increased serum TNF-α levels positively correlate with increased anxiety and/or depression [56]. In this systematic review, we were unable to find any evidence of significant variations in systemic TNF-α levels between concussed and non-concussed groups having emotional symptoms. However, we found that elevated TNF-α levels within mTBI population is associated with adverse psychological outcome(s) [37,38]. In conclusion, elevated TNF-α levels are associated with PTSD in chronic stages especially in the male-predominant mTBI population.

IL-10 is a prominent anti-inflammatory cytokine detected in both acute and chronic stages of mTBI [20,37]. In the absence of a head injury, the relation between IL-10 and psychological outcomes, PTSD, and depression is inconclusive [57,58]. For example, one study demonstrated reduced serum IL-10 levels in depressed patients in comparison to non-depressed patients [59], though other studies identified upregulation in serum IL-10 level within depressed patients [59,60,61]. Similarly, studies have found both significantly elevated and depressed IL-10 levels in PTSD patients in comparison to control groups [62,63,64]. Overall, this review supports the finding that elevated IL-10 levels seen in chronic mTBI patients are associated with PTSD and depression. This could be seen as a compensatory mechanism for increased inflammation observed following a concussion [26].

CRP is a non-specific marker of systemic inflammation. Following a mTBI event, CRP levels increase for several days before gradually declining, potentially over weeks [65]. Elevated levels of serum CRP, specifically within the first 24 h of a mTBI event, are associated with greater injury severity [36]. High-sensitivity serum CRP levels within two weeks of an mTBI event are prognostic biomarkers for potential disability six months later [65,66]. Elevated CRP levels are also seen to be associated with psychological impairments, even in the absence of a mTBI event [67]. Patients with PTSD exhibit significantly upregulated CRP levels in comparison to those who did not meet the clinical criteria for PTSD [68,69]. Similarly, patients with depression and anxiety have upregulated CRP levels in comparison to controls [67,70,71]. Overall from this review, it is evident that elevated CRP levels in the mTBI patient population are positively associated with depression and PTSD. Additional research is required to further explore this correlation.

IL-1β levels are acutely elevated following a concussion [19,20]. Elevated IL-1β levels are frequently associated with emotional symptoms even without mTBI [72,73]. Based on the current evidence, acutely elevated IL-1β levels are associated with adverse emotional outcomes in mTBI patients. However, more studies are needed to further support this.

## 5. Limitations

The limitations of this systematic review arise from the quality of the evidence available regarding this topic, as all the included studies had level IV evidence. The study design, comparative groups, biomarkers investigated, and variations in psychological assessments contribute to heterogeneity. Moreover, incomplete documentation of data (for example, specific levels of biomarkers, psychological scores, and certain outcomes) for the population of interest limited our ability to ascertain the influence of several biomarkers on specific psychological outcomes.

Medications, autoimmune disorders, and other comorbidities (e.g., metabolic syndrome, type II diabetes) affecting the immune system are some of the factors that can impact the levels of cytokines within the investigated patient populations. Although some studies included in this review accounted for these factors through their exclusion criteria, most of the studies did not.

A very large percentage of the study population, across all included studies, were male. Future studies should aim to include more female participants so that the samples can be more representative of the mTBI population.

Furthermore, there are variations between the included studies regarding the time of cytokine assessment post-injury. Most studies included assessed participants in the chronic stage of PCS (from one month and up to multiple years), and it is not clear what elevations of cytokines in the acute stage of mTBI are associated with increased emotional symptoms. The specific times of sample collection post-injury should be reported in future studies. In addition, the scales used to assess the psychological outcomes should be more standardized in order to reduce data heterogeneity.

Despite these limitations, the datasets from this study do support an association between emotional symptoms and neuroinflammation in the concussed population.

## 6. Conclusions

There is a positive correlation between elevated IL-6 and TNF-α levels and PTSD in chronic mTBI patient population. Similarly, elevated IL-10 and CRP levels are associated with PTSD and depression in chronic mTBI population. However, it is challenging to ascertain a clear relationship between individual inflammatory cytokines and specific psychological outcomes in mTBI population, due to the heterogeneity in biospecimens analyzed, time of cytokine assessment post-injury, outcome assessments, comparative groups, lack of female participants, and control for confounding factors. Additionally, more standardized protocols for cytokine and psychological outcome analysis should be utilized in future mTBI studies to allow for a clearer understanding of the relationships studied. Measurement of these cytokines can prove useful as biomarkers to identify patients at risk of emotional symptoms following a mTBI event, prompting early rehabilitation and hence expediting patient recovery.

## 7. Clinical Significance

Chronically elevated IL-6 and TNFα levels following mTBI could be used to identify patients at risk of developing PTSD. Similarly, chronically elevated IL-10 and CRP levels could help identify mTBI patients at risk of developing depression and PTSD. Early detection of patient populations at risk of poor emotional outcome(s) would help clinicians plan early rehabilitation to mitigate losses.

## Figures and Tables

**Figure 1 brainsci-12-00102-f001:**
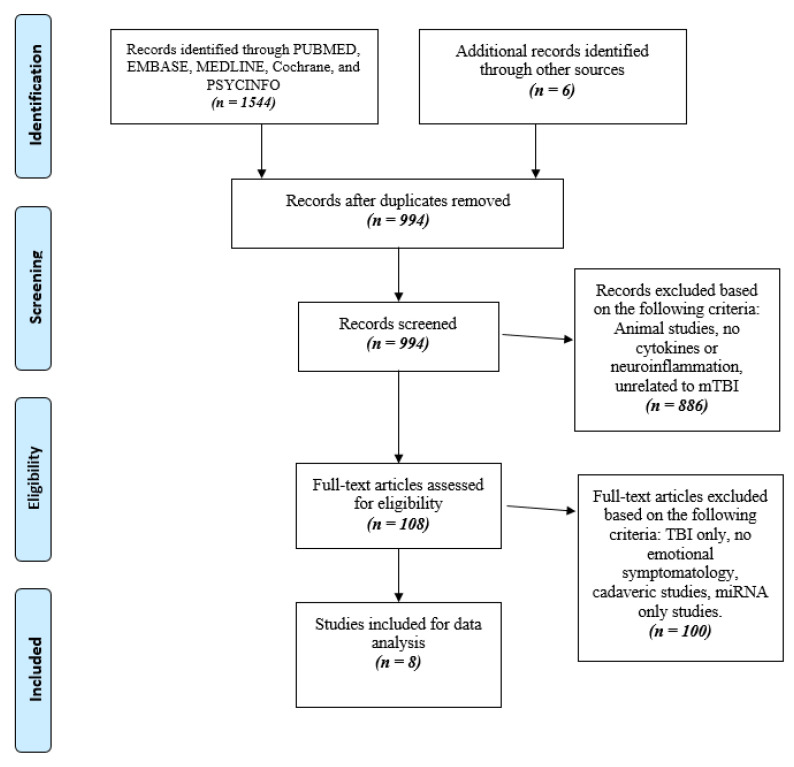
Prisma flow chart detailing the stages of study retrieval and selection.

**Table 1 brainsci-12-00102-t001:** Study characteristics and outcomes.

Author and Year	Population	mTBI Setting	Cytokine(s)	Biospecimen	Outcome of Interest	Acute/Chronic (Average Time)
Bellgowan et al. (2012)	*n* = 9 (Cases = 9, Controls = 0)	Sports	IL-1β	Blood—venous plasma	IL-1β levels were significantly elevated at 24–48 h compared to either 1 week or 1 month post-concussion. Depression and anxiety ratings significantly higher at 24–48 h and 1 week, but not 1 month after concussion.	Acute (24 h, 1 week–4 weeks)
Ghai et al. (2020)	*n* = 69 (Cases = 27, Non-mTBI Veterans (DC) = 11, Community Controls (CC) = 31)	Military	CRP	Blood—venous plasma	Elevated CRP levels in chronic mTBI patients compared to CC and DC controls. Chronic mTBI group had significantly greater comorbid PTSD and depressive symptoms, compared to CC and DC.	Chronic (4.6 years)
Gill et al. (2018)	*n* = 64 (Cases = 42, Controls = 22)	Military	IL-6, IL-10, TNFα	Blood exosomal plasma	Chronic mTBI patients had elevated exosomal IL-10 levels compared to controls. PTSD was significantly related, and depression tended to be related to exosomal IL-10.	Chronic (3–36 months)
Guedes et al. (2020)	*n* = 195 (Cases = 150, Controls = 45)	Military	IL-6, TNF-α, IL-10	Blood venous plasma	mTBI group had increased PTSD and depression symptoms compared to controls. Plasma TNF-α (*p* = 0.02) and exosomal IL-6 (*p* = 0.08) levels correlated with PTSD.	Chronic (NR)
Kanefsky et al. (2019)	*n* = 143 mTBI+LOC (*n* = 25), mTBI without LOC (*n* = 36), Controls = 82)	Military	TNF-a, IL-6, IL-10	Blood venous plasma	Both mTBI groups (+/− LOC) reported significantly greater depression and PTSD symptoms compared to controls. IL-6 was elevated in the mTBI with LOC group compared to both the mTBI w/out LOC and control groups. Within the mTBI groups, increased TNF-α concentrations were associated with greater PTSD symptoms (*r* = 0.36, *p* = 0.005).	Chronic (NR)
Peskind et al. (2015)	*n* = 105 (Cases = 35, Non-mTBI Veterans (DC) = 16 Community Controls (CC) = 55	Military	IL-7, IL-6	CSF	mTBI veterans had greater PTSD and depression and elevated CSF IL-7 levels compared to the DC. CSF IL-6 levels did not differ between mTBI Veterans and DC but was significantly higher in both Veteran groups than in CCs.	Chronic (NR)
Su et al. (2014)	*n* = 213 (Cases = 213, Controls = 0)	Trauma	CRP	Blood venous plasma	CRP levels were significantly correlated with PCS, persistent psychological problems and persistent cognitive impairments.	Chronic (1–3 months)
Vedantam et al. (2021)	*n* = 77 (Cases = 53, OI Controls = 24)	Trauma	IL-1β, IL-2, IL-4, IL-6, IL-10, IL-17, IFN-γ, TNFα, IL-12p70	Blood exosomal plasma	Within 24 h, IL-2 and IL-6 levels were significantly elevated in the mTBI population vs. OI controls. At 6 months post-injury, mTBI group had elevated IL-6 (*p* = 0.044) levels vs. OI controls. Elevated IL-10 levels at 6-month post-mTBI was significantly associated with severe PTSD (*p* = 0.004) symptoms and worse mood (*p* = 0.001).	Acute (24 h)/ Chronic (6 months)

Abbreviations: mTBI, mild traumatic brain injury; IL, Interleukin; DC, deployed controls (non-mTBI Veterans); CC, community controls; CRP, C-reactive protein; PTSD, post-traumatic stress disorder; TNF, tumor necrosis factor; NR, not reported; LOC, Loss of Consciousness; CSF, cerebrospinal fluid; PCS, post-concussive symptoms; OI, orthopedic injury; IFN, interferon; Definitions: Acute refers to a concussion that was diagnosed less than 4 weeks ago. Chronic refers to a concussion that was diagnosed more than 4 weeks ago.

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
