# Peer review of "Correlation between Mild Traumatic Brain Injury-Induced Inflammatory Cytokines and Emotional Symptom Traits: A Systematic Review"

_brainsci, 2022, doi:10.3390/brainsci12010102_

Round 1

Reviewer 1 Report

The authors present a manuscript concerning the changes in pro(and anti) inflammatory cytokines in patients with mTBI. This is a review article which incorporates only 8 studies in the final analysis. This is a topic which has significant implications for long-term disability and may prove to be of great use in the future as medical and trauma care becomes more individually tailored. The authors are to be commended for their work in helping to create a single review of potential targets for study. 

Intro: No significant issues. Could expand a little on the specific PCS symptoms and whether or not prior studies associated any particular cytokines with them.

Clinical Significance: Is there a time when the elevated levels become chronic? 

Overall a well-written and interesting manuscript. Just some very minor changes.

Author Response

Point 1: The authors present a manuscript concerning the changes in pro(and anti) inflammatory cytokines in patients with mTBI. This is a review article which incorporates only 8 studies in the final analysis. This is a topic which has significant implications for long-term disability and may prove to be of great use in the future as medical and trauma care becomes more individually tailored. The authors are to be commended for their work in helping to create a single review of potential targets for study. 

Intro: No significant issues. Could expand a little on the specific PCS symptoms and whether or not prior studies associated any particular cytokines with them.

Clinical Significance: Is there a time when the elevated levels become chronic? 

Overall a well-written and interesting manuscript. Just some very minor changes.

Response 1:

Thank you for the feedback. We have added a few lines demonstrating the association of inflammatory cytokines with common symptoms such as headaches, cognitive impairments, sleep disturbances, and emotional impairments. (Line 50-58)

Regarding chronic cytokine levels, it is difficult to pinpoint the absolute time-period beyond which elevated post-mTBI cytokine levels are considered chronic. Based on the current evidence, it is observed that cytokine levels return to baseline within hours to a few days. However, if the cytokines levels fail to return to the baseline and remain elevated in the blood. for weeks and months, they are generally considered chronically elevated.

Reviewer 2 Report

The title excludes the word cytokines and relies solely on “neuroinflammation” which can imply a myriad of pathways and mechanisms unrelated to inflammatory cytokines. Consider incorporating the use of “inflammatory cytokines” into the title.

Line 123-out of 994 papers only 8 met the inclusion criteria. The rationale not to include moderate or severe TBI articles in this review should be addressed in the manuscript. Was the exclusion of studies with GCS <13 a major reason for study exclusion? As part of the limitation section, it might be beneficial to note you’re your findings may or may not be generalizable to moderate or severe TBI.

Line 83 -Emotional symptomatology was an inclusion criteria, however Lines 193-194 mention that 5/8 were patients that also presented with emotional symptoms.  Shouldn’t all 8 studies included express emotional symptoms? The same comment applies to lines 220-221. Please clarify.

Can time of serum samples after mTBI be included in Table 1. Perhaps under chronic put the range of years, as it would be beneficial to know whether we are speaking of less than 5 years or 25 years. I recognize that it might not be reported in all 8 articles, those could be denoted as post-TBI time frame (NR) “not reported”

Lines 287, 295, 355 and elsewhere, the symbol for beta is not consistent with Table 1 and other locations throughout the manuscript. The symbol used in Table 1 is preferred.

Line 324- Sentence should begin with the word “Elevated” or “Increased”

Lines 356-359. The use of “may be inferred” seems somewhat weak if the goal was to assess the level of evidence. It would appear to me that the level of current evidence would either support or be insufficient at this time to support an association between an elevated cytokine and mTBI with emotional symptomology. This wording of may be inferred is used multiple times, consider clarifying or rewording this phrase, perhaps something of the nature, “Based on the current evidence elevated IL-1B levels are or are not associated with…..”

Author Response

Point 1: The title excludes the word cytokines and relies solely on “neuroinflammation” which can imply a myriad of pathways and mechanisms unrelated to inflammatory cytokines. Consider incorporating the use of “inflammatory cytokines” into the title. 

Response 1: The title has been modified to “Correlation between Mild Traumatic Brain Injury-Induced Inflammatory Cytokines and Emotional Symptom Traits: A Systematic Review”

Point 2: Line 123-out of 994 papers only 8 met the inclusion criteria. The rationale not to include moderate or severe TBI articles in this review should be addressed in the manuscript. Was the exclusion of studies with GCS <13 a major reason for study exclusion? As part of the limitation section, it might be beneficial to note you’re your findings may or may not be generalizable to moderate or severe TBI.

Response 2: We accept the reviewers point that this review may not be generalizable to moderate or severe TBI patients, however it is applicable to the majority of the TBI population as mild TBI or concussion constitutes more than 90% of the all TBIs. Additionally, emotional issues are more predominant in the mTBI group in comparison to other more severe forms of TBI. A rationale for only including mTBIs has been added in lines 71-73. GCS < 13 was a major reason for study exclusion because this review only focuses on mTBIs, where GCS > 13. (Line 90)

Point 3: Line 83 -Emotional symptomatology was an inclusion criteria, however Lines 193-194 mention that 5/8 were patients that also presented with emotional symptoms.  Shouldn’t all 8 studies included express emotional symptoms? The same comment applies to lines 220-221. Please clarify.

Response 3: The inclusion criteria was modified from “emotional symptomatology” to “emotional symptomatology (anxiety, depression or PTSD)”, to accommodate any study that discusses either of the three selected emotional conditions (Line 88). All eight included studies discuss emotional symptoms, so lines 193-194 (now Line 203-204) and 220-221 (now Line 230-231) were changed to only mention how many of the included studies discuss the inflammatory cytokine of interest, ex. Line 203-204 now says “Five of the eight included studies assessed IL-6 levels within the mTBI population”.

Point 4: Can time of serum samples after mTBI be included in Table 1. Perhaps under chronic put the range of years, as it would be beneficial to know whether we are speaking of less than 5 years or 25 years. I recognize that it might not be reported in all 8 articles, those could be denoted as post-TBI time frame (NR) “not reported”

Response 4: The 7th column of Table 1 will now include the average time of serum collection, following the injury. Additionally, chronic and acute were defined in section 2.3 Data Abstraction, under Methods, and a definition will now be included under Table 1 as well.

Point 5: Lines 287, 295, 355 and elsewhere, the symbol for beta is not consistent with Table 1 and other locations throughout the manuscript. The symbol used in Table 1 is preferred.

Response 5: All symbols are now consistent with the ones presented in Table 1.

Point 6: Line 324- Sentence should begin with the word “Elevated” or “Increased”

Response 6: The sentence was changed to “Elevated TNF-α levels are associated with poor psychological health even in the absence of an mTBI event”. (Now Line 333)

Point 7: Lines 356-359. The use of “may be inferred” seems somewhat weak if the goal was to assess the level of evidence. It would appear to me that the level of current evidence would either support or be insufficient at this time to support an association between an elevated cytokine and mTBI with emotional symptomology. This wording of may be inferred is used multiple times, consider clarifying or rewording this phrase, perhaps something of the nature, “Based on the current evidence elevated IL-1B levels are or are not associated with…..”

 Response 7: 

We agree with the reviewer. Following corrections were made:

  1. Overall, based on the available literature, elevated IL-6 levels are associated with PTSD in the chronic mTBI population. (Now Line 326-328)
  2. In conclusion, elevated TNF-α levels are associated with PTSD in chronic stages especially in the male predominant mTBI population. (Now Line 340-342)
  3. Overall, from this review, it is evident that elevated CRP levels in the mTBI patient population are positively associated with depression and PTSD. (Now Line 362-364).
  4. Based on the current evidence, acutely elevated IL-1ß levels are associated with adverse emotional outcomes in mTBI patients. However, more studies are needed to further support this. (Now Line 366-368).